



# A climate-conditioned catastrophe risk model for UK flooding

Paul D. Bates[1,2], James Savage[2], Oliver Wing[1,2], Niall Quinn[2], Christopher Sampson[2], Jeffrey Neal[1,2] and Andrew Smith[2]

[1]School of Geographical Sciences, University of Bristol, Bristol BS8 1SS, UK
[2]Fathom, Square Works, 17-18 Berkeley Square, Bristol, BS8 1HB, UK

*Correspondence to*: Paul Bates (paul.bates@bristol.ac.uk)

**Abstract.** We present a climate-conditioned catastrophe flood model for the UK that simulates pluvial, fluvial and coastal flood risks at 1 arc second spatial resolution (~20-25m). Hazard layers for ten different return periods are produced over the whole UK for historic, 2020, 2030, 2050 and 2070 conditions using the UKCP18 climate simulations. From these, monetary 
losses are computed for Great Britain only for five specific global warming levels (0.6, 1.1, 1.8, 2.5 and 3.3°C). The analysis contains a greater level of detail and nuance compared to previous work and represents our current best understanding of the UK's changing flood risk landscape. Validation against historical national return period flood maps yielded Critical Success Index values of 0.65 and 0.76 for England and Wales respectively, and maximum water levels for the Carlisle 2005 flood were replicated to an RMSE of 0.41m without calibration. This level of skill is similar to local modelling with site specific data. 
Expected Annual Damage in 2020 was £730M, which compares favourably to the observed value of £714M reported by the Association of British Insurers. Previous UK flood loss estimates based on government data are ~3x higher and lie well outside our modelled loss distribution, which is plausibly centred on the observations. We estimate that UK 1% annual probability flood losses were ~6% greater in the average climate conditions of 2020 than for the period of historical river flow and rainfall observations (centred approximately on 1995) and can be kept to around ~8% if all countries' COP26 2030 carbon emission 
reduction pledges and 'net zero' commitments are implemented in full. Implementing only the COP26 pledges increases UK 1% annual probability flood losses by ~23% above recent historical values, and potentially ~37% if climate sensitivity turns out to be higher than currently thought.

## 1 Introduction

Flooding is the principal environmental hazard identified in the UK's National Risk Register (Cabinet Office, 2020), and past 
major events have resulted in substantial economic damage and loss of life. For example, the coastal floods of 1953 resulted in 307 deaths, whilst inland flooding during summer 2007 inundated around 55,000 homes and left more than 400,000 people temporarily without drinking water. Despite significant investment in river and coastal defences over the last fifty years, including a further £5.2Bn to be invested from 2021-2026 (Cabinet Office, 2020), floods continue to be a problem for the UK, with major events occurring in winters 2013-14, 2015-16 and 2019-20 and in summer 2021. In England alone, the 
Environment Agency has previously stated that over 5M properties, or around 1 in 6 of the total building stock, have a greater





than 0.1% annual probability of either fluvial or coastal flooding or an unspecified probability of pluvial flooding (Environment Agency, 2009). Flood risks in the UK will also very likely increase in the future as a result of population growth, changes in vulnerability and anthropogenic climate change (Merz et al., 2021; The Committee for Climate Change, 2021)

The reasons for this are not difficult to see. The UK lies under the westerly track of mid-Atlantic storm systems (including
extratropical cyclones) that can cause storm surges and extreme waves on exposed coasts (Haigh et al., 2017). On making landfall, these storm systems encounter extensive upland areas to the west of the country resulting in orographic enhancement of precipitation. This subsequently falls onto river catchments that are (in global terms at least) relatively short and steep and therefore prone to flooding (Black and Law, 2004; Luca et al., 2017). Convective rainfall activity in summer can be intense (Chan et al., 2016) and may lead to flash flooding in urban areas and small catchments (Archer and Fowler, 2021), whilst
atmospheric rivers can cause major flood events during winter (Lavers et al., 2011). Along the eastern seaboard lies the shallow marine basin of the North Sea, which is effectively closed at its southern end by the Straits of Dover and is therefore a setting extremely conducive to the development of storm surge flooding (Horsburgh and Wilson, 2007). Moreover, the UK is densely populated (281 people/km$^2$, 67.2M total population in 2020) with development often concentrated in low-lying and flood prone areas along the coasts and major rivers.

Despite the threat posed by floods, the methods currently used to map national scale flood hazard and risk in the UK are, at best, opaque. The approaches adopted by government bodies and commercial organisations are largely undocumented, either in the peer-reviewed literature or in accessible reports, and validation studies are rarely reported. Indeed, considerable detective work is required to even understand what methods and data sets underpin existing flood risk information in the UK, despite these being used to inform critical long term planning appraisals such as the UK's Climate Change Risk Assessment
(The Committee for Climate Change, 2021) or the national level of investment in flood defences (Environment Agency, 2019b). This lack of transparency is, from a scientific standpoint, unhelpful and unhealthy, and likely to hinder robust decision making. In addition, data sets available for flood research either represent only a limited range of return periods, do not consider the spatial correlation in flood hazard (c.f. Heffernan & Tawn, 2004; Keef et al., 2009, 2012) or do not account for climate change impacts.

The purpose of this paper is therefore to address both technical and transparency issues in the flood risk information available in the UK. Neither is the UK unique in this respect: official flood mapping approaches in many countries lack scientific transparency and accountability (e.g. Pralle, 2019), and the methods are rarely subject to peer-review. Lessons learned through this work are therefore likely to be more widely applicable, for example for the Federal Emergency Management Agency flood mapping programme in the US and for modelling conducted in support of the European Floods Directive in the European
Union. Accordingly, we describe the development and validation of a climate-conditioned catastrophe risk model for UK pluvial, fluvial and coastal flooding. Current UK flood hazard and risk data sets are reviewed in Section 2. The methodology underpinning the model is described in detail in Section 3. Validation results and projections of current and future risk are presented in Section 4 and conclusions drawn in Section 5. Details of how to obtain the data for academic use are given at the end of the paper.



## 2 Current UK flood hazard and risk datasets

UK flood hazard and risk information at national and sub-national scales can be found in five broad classes of data product. A brief description of each follows, with further information in Section S1.

### 2.1 UK flood hazard maps

Floodplain zonation (i.e., hazard) maps for fluvial, coastal and sometimes pluvial flooding are developed separately by government bodies in the devolved regions of the UK (Northern Ireland, Wales, Scotland and England) and predominantly used to inform land use planning decisions. These maps are often constructed using a patchwork of local modelling studies commissioned from commercial engineering consultants for individual river reaches. In England alone there are over 2000 local hydraulic models that have been developed in this way to map the 1 in 100 year fluvial floodplain, with inundation in unmodeled areas likely to have been filled in from historic observations or local knowledge. The process is therefore similar to that used by the Federal Emergency Management Agency (FEMA) in the US (see Wing et al., 2018) and other hazard management organisations worldwide. The individual reach scale modelling studies from which the national map is derived typically use 1D, 1D/2D or 2D hydraulic models with airborne LiDAR floodplain elevation data, bathymetric survey information and most commonly represent undefended conditions. For England, a 1 in 1000 year floodplain layer was also created by a 50m spatial resolution national scale hydraulic model developed in the early 2000s (Bradbrook et al., 2005) and pluvial flooding has also been mapped nationally. For coastal flooding, simple GIS-based 'bathtub' models, which can severely overestimate areas at risk (Vousdoukas et al., 2016), may be used instead of true hydrodynamic simulations. Model boundary conditions are usually derived from either extreme value frequency analysis of long duration river, tide or meteorological records or from a UK standard regionalised flood frequency approach for discharge in ungauged basins in the case of river flows (Robson and Reed, 1999). Where available, model outputs may be calibrated to match observations of historic floods. The resulting flood maps therefore represent average conditions over the period of the instrumental record in the UK (typically from the 1960s onwards at most sites). The true present-day hazard will therefore differ from the recent historic average because of natural climate variability and already-observed (but modest) changes in extreme rainfall resulting from anthropogenic climate change (Kendon et al., 2021).

Beyond this broad overview, obtaining more detailed information to properly understand how UK flood hazard mapping is conducted is extremely difficult. Some small streams are not covered by this mapping, and it is clear from national water body datasets that catchment headwater areas can be missed out. Despite this, no metadata exists showing the spatial limit of the dataset so we cannot tell the difference between areas with no flood risk and those simply with no data because they have not been modelled. No information is provided on the individual local models, the specific data sets used, the model simulation performance or when the modelling was completed. This latter piece of metadata is important because the complete national map required the commissioning of thousands of individual local studies and therefore took a significant amount of time to complete. As a result, some of the national map's component models are likely to be outdated. The strength of these flood



hazard maps is that they are often created using well established hydraulic modelling approaches by trained engineers who have good access to local data. For this reason, and because of their official status, they are typically considered the 'gold standard' in flood modelling, however no systematic assessment of such hazard mapping has ever been presented.

## 100  2.2 UK flood risk maps

Flood risk maps or spatially aggregated risk data (i.e., the product of flood probability, exposure and vulnerability) are also produced by the devolved administrations and are predominantly used to inform flood defence investment policy and long-term risk planning. Risk is quantified either in terms of the number of properties exposed to flooding of a given probability or the Expected Annual Damage (EAD). More formally, the latter quantity is the integral of the loss-exceedance probability
curve for a particular hazard. Data are presented as economic losses, so represent the current value of assets that are damaged by the flood event minus any taxation element.

Only limited information on how this is done is available publicly, but it is apparent that the four administrations vary in terms of the approach adopted, the flood probabilities which are reported and which sources of flooding these represent (see Table S1). Most information is available in England, where flood risk maps are produced by the Environment Agency as part of
their National Flood Risk Assessment (NaFRA) programme (Environment Agency, 2009). A summary of what is known of the method is provided in Section S1.2. In Scotland and Wales no information on the method used is made available and only the total number of properties exposed to flooding is openly reported. In Northern Ireland, it appears that a simple GIS overlay of flood hazard maps and exposure has been undertaken to calculate risk, but no further details are in the public domain. Spatial correlations in flood depths (c.f. Heffernan & Tawn, 2004; Keef et al., 2009, 2012; Quinn et al., 2019) are not taken
into account by any of these methods so only average annual losses can be computed and not the full loss-exceedance curve.

No public validation of these risk outputs has been undertaken by the government agencies responsible, but Penning-Rowsell (2014) and Penning-Rowsell (2021) have shown how the methods and output from the NaFRA analysis in England have changed significantly over time. In particular, the raw output from NaFRA 2008 indicated implausibly high flood losses (EAD of >£5Bn, see Table S1). This was determined to be the result of excessive predicted floodplain water depths and Penning-
Rowsell (2021) has documented the significant and somewhat arbitrary adjustments that have been made since 2008 to try to combat this. These changes are described in more detail in Section S1.3 and include switching to a simpler loss calculation that does not use water depth as an input, capping losses for low return period events, limiting floodplain water levels and manual adjustment of losses in areas where the results are deemed implausible. By 2018 these and other methodological changes had reduced the EAD in NaFRA to £0.66Bn, but even so this value was still 2-9x higher than comparable loss data
from the Association of British Insurers depending on the assumptions made (Penning-Rowsell, 2021; see also Section 2.5 below and Table S1).



## 2.3 Current and future flood risk estimates produced as part of the UK's Climate Change Risk Assessment (CCRA) process.

Changes in flood risk as a result of climate change are obviously an important consideration for policy makers, and this requires a consistent UK-wide analysis. Given the differences in flood risk assessment methods and reporting between the devolved regions of the UK outlined in Sections 2.1 and 2.2 this is simply not possible using the data sets described above. Instead, the Future Flood Explorer methodology of Sayers et al. (2016) and Sayers (2017) is used in the UK's five yearly cycle of Climate Change Risk Assessment (The Committee for Climate Change, 2021) to address this limitation. Future Flood Explorer is a statistical emulation approach to fill gaps in the existing hazard and risk information available from the responsible authorities
in the UK's devolved regions (see Section S1.3 for further details). This spatially consistent information can then be extrapolated into the future allowing for different climate, socio-economic and adaptation scenarios. However, the method necessarily inherits any of the errors in the underlying hazard and risk data sets produced by the devolved administrations and unsurprisingly produces similar results for EAD (see Table S1). Similar to NaFRA, the loss calculation in the method does not use depth-damage curves, and spatial correlations in flooding between locations are also not accounted for. As a result,
only an Expected Annual Damage can be computed and not the full loss exceedance curve. Validation of the outputs has been undertaken for the number of properties flooded during the 2007 summer floods in England which showed a 2.2x over-estimation by the Future Flood Explorer versus the best available post-event reconstruction (Sayers, 2017), consistent with the results for NaFRA of Penning-Rowsell (2021).

## 2.4 Flood hazard and risk data produced by commercial modelling firms

A number of catastrophe risk modelling firms produce stochastic models of UK flooding on behalf of the insurance industry and other sectors. However, the methods and data from these schemes are typically regarded as commercially sensitive and few details are available in the public domain. To date, there are no peer-reviewed journal publications or comprehensive public validation studies for the UK instances of these approaches and the data are not available for academic use. Anecdotally we know that, unlike the publicly available data above, these methods do take flood spatial dependence into account and so
can provide the full loss-exceedance curve. Underlying hazard data are available for a range of return periods and climate change scenarios may also have been computed for some of the models.

## 2.5 Flood losses recorded by the Association of British Insurers

Finally, the Association of British Insurers (ABI) collates data on payouts by its members in respect of residential flooding claims. These data have been produced annually since 1998 and are described in detail in Penning-Rowsell (2014) and
Penning-Rowsell (2021). Unlike UK Government risk data, the figures represent total financial losses (i.e., the actual money paid out) due to flooding from all sources for the whole of the UK. The figure does not account for losses incurred by the ~20% of insurers who are not members of the ABI or for under-insurance by householders. Further, as Penning-Rowsell

(2014) explains, insurance policies in the UK are written on a 'new-for-old' replacement basis such that each payout includes an element of 'betterment' (the difference between the cost of a new item and the second-hand value of the flood damaged one it replaces). To convert financial to economic losses, this betterment element and any taxation (e.g., sales tax) needs to be deducted (see Section S2), and adjustments are required for ABI market share and homeowner under-insurance. Despite these caveats, the ABI data do provide a set of observed annual flood losses to compare to modelled estimates.

## 2.6 Summary

It is clear from the above review and further details in Section S1 that important details of the methods and data used to create current flood risk products for the UK are not available in the public domain. Despite this, a lack of alternatives means that these data necessarily underpin nearly all current academic studies of flood risk in the UK (e.g. Rözer & Surminski, 2021; Sayers et al., 2018). Significant inconsistencies occur between the different devolved regions of the UK and most only represent historic average conditions rather than the present day or future. Validation is limited, and those data sets that are publicly available cannot be used to answer important scientific questions about extreme UK annual flood losses and the impacts of climate change.

## 3. Methods

To address the challenge identified above, we here describe a climate-conditioned catastrophe risk model for UK pluvial, fluvial and coastal flooding for historic, current and future conditions. A detailed description is provided in S3 and a summary in Figure 1. The method can broadly be conceptualised by as: (i) the creation of hazard layers across the UK for specified return period intervals and climate scenarios; (ii) the characterisation of spatial dependence in flood footprints and synthetic event catalogue generation through sampling from the existing hazard layers; and finally, (iii) the intersection of exposure data with vulnerability loss functions and event depths to estimate loss.



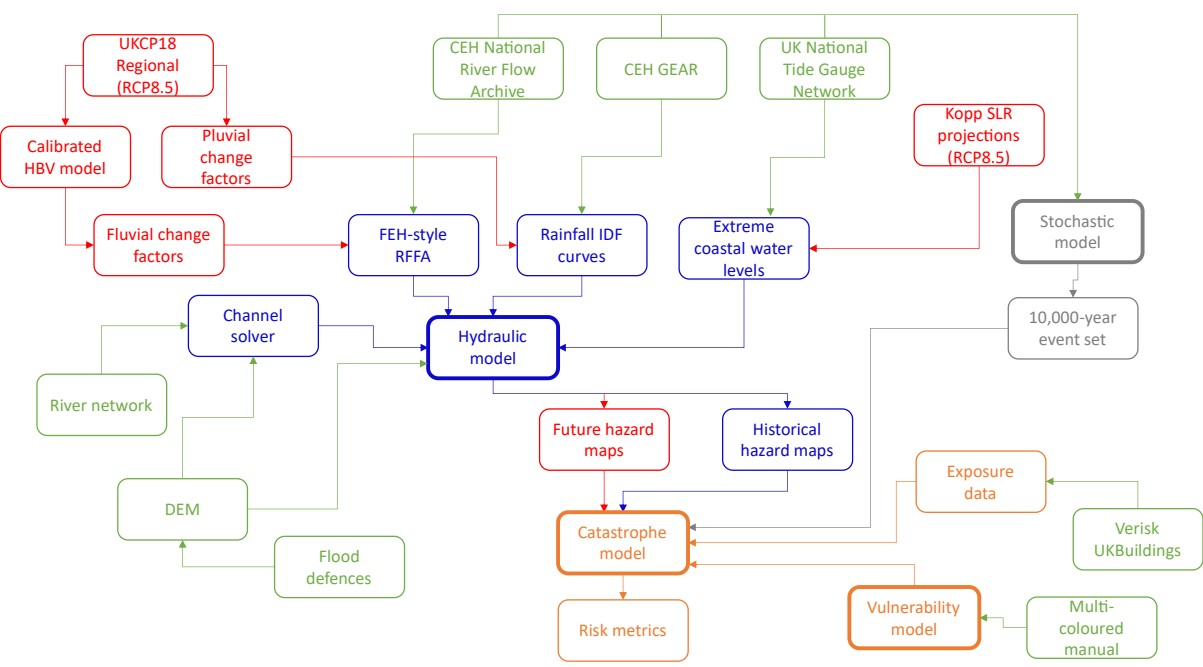

**Figure 1: Method employed to compute climate-conditioned UK flood hazard and risk maps using a catastrophe model approach.**
**See sections S3.1 to S3.3 for further details.**

At the heart of the method (blue cells in Figure 1) lies a standard 1D/2D hydraulic model. In our case this is a variant of the LISFLOOD-FP code (Almeida et al., 2012; Almeida & Bates, 2013; Bates et al., 2010; blue, in bold in Figure 1). However, any comparable model would give similar results (c.f. Hunter et al., 2008). The 2D component of the model is run over the whole of the UK at 1 arc second spatial resolution (~20-25m at this latitude). Floodplain elevation values are derived from a

composite Digital Terrain Model (DTM) built using ~ 15cm vertical accuracy LiDAR data and UK Ordnance Survey terrain data where LiDAR does not exist. LiDAR coverage is approximately 70% by area and concentrated in lowland zones. River channels are treated as 1D sub-grid scale features following Neal et al. (2012) so are not constrained by the resolution of the 2D model and can be of any width. Channel locations are defined using UK Ordnance Survey channel location data and widths are derived based on empirical relationships with upstream catchment accumulation area i.e., a hydraulic geometry approach.

Rectangular channels are assumed, and effective channel bed elevations are estimated using the methods of Neal et al. (2021). This is an optimization approach which ensures that the channels are appropriately sized for the flows being simulated, mitigating the problem of gross mismatches between discharge and channel conveyance and any approximations made when estimating the channel width. We assume a bankfull return period of 1 in 2 years for natural alluvial channels and the appropriate standard of protection along defended reaches. Flood defence information is taken from the data made available

by UK government agencies (AIMS Spatial Flood Defences (inc. standardised attributes), 2022), supplemented by a levee detection algorithm to fill in any gaps (Wing et al., 2019). Friction parameters in the model are as standard values across the UK.



Blue cells in Figure 1 also denote the boundary conditions for the hydraulic model and its main outputs, which are a series flood hazard maps for ten different return periods from 1 in 5 to 1 in 1000 years for historical conditions. Boundary conditions

for fluvial, pluvial and coastal floods are obtained from, respectively, a Regionalised Flood Frequency Analysis of UK gauged flows from the National River Flows Archive using an FEH-type method (Robson and Reed, 1999), rainfall Intensity-Duration-Frequency curves derived from the CEH-GEAR1h database (Lewis et al., 2018) and the likelihood of coastal extreme water levels derived from the UK tide gauge network (Environment Agency, 2019a). Data inputs to this process are shown as green cells in Figure 1. The baseline data therefore represent the extreme value distribution calculated over the period of the historical

record (approximately 1960 to the present day for river flow and sea level and 1990-2014 for rainfall), with sea level de-trended based on 2018 mean sea level values. The historic extreme event return periods represent an average of the observation period, acknowledging that each gauge site has a different record length which makes a precise evaluation difficult. Given changes in the number of stations over time (see https://nrfa.ceh.ac.uk/uk-gauging-station-network for the river gauge network) this is somewhere around 1990-2000, which coincidentally is also when the planet reached 0.6°C of warming above pre-

industrial conditions (Hausfather and Moore, 2022).

These historic boundary conditions are then adjusted to current (2020) and future (2030, 2050, 2070) conditions using future climate projections from the UKCP18 12km Regional simulations under the RCP8.5 scenario and sea level rise projections from Kopp et al. (2014). The UKCP18 projections shows no significant change in storm surge so we assume the UK storm surge climate (as captured in tide gauge data) persists into the future. Changes in extreme precipitation and sea level are used

directly in the modelling, but for changes in extreme river discharge the future rainfall is used as input to a set of ~1000 catchment hydrology models that are used to compute river flow change factors. Parameter uncertainty is accounted for in the hydrological simulations and the results are regionalised to give full national coverage. These change factors (see Figure S2) are then used to produce future flood hazard maps as denoted by the red cells in Figure 1. Lastly, stochastic modelling is used to generate realistic event footprints by characterizing the spatial dependence in flooding (grey cells in Figure 1). The spatial

dependence is determined using a conditional exceedance statistical model (Heffernan & Tawn, 2004; Keef et al., 2009, 2012) and this information is then used to sample synthetic events from the pre-computed hazard layers (c.f. Quinn et al., 2019). By combining these hazard event footprints with exposure data and vulnerability functions we are able to compute financial losses and obtain the full loss-exceedance probability distribution. Whilst the hazard model is run over the whole of the UK, suitable exposure data are not publicly available over the whole of Northern Ireland so for now loss computations are restricted to

Great Britain (Wales, Scotland and England). To determine exposure, we use the Verisk UKBuildings (formerly 'Geomni') data set available for academic use through Digimap (https://digimap.edina.ac.uk/verisk) which gives information on property type, age and use for each building in the Great Britain and for vulnerability we use a slightly modified set of standard UK depth-damage curves (the so called Multi-Coloured Manual approach; see Penning-Rowsell et al., 2013). Finally, loss results are presented in terms of specific global warming levels to decouple these from the specific Representative Concentration

Pathway, RCP8.5, simulated by UKCP18 Regional.



## 4. Results and discussion

### 4.1 Model validation

Outputs from the hazard model were first compared to equivalent return period flood extent maps produced by the devolved administrations in the UK (see Section 2.1 and S1.1). Specifically, we compared the 1 in 100-year return period hazard layer produced by the historic run of the national model to the complete set of equivalent flood hazard maps produced by the Environment Agency for England and Natural Resources Wales using (mostly) a patchwork of local models. Whilst this is a model-to-model comparison where neither simulation represents truth, we assume that the local models can potentially have higher skill because they have been built manually using local data and are typically calibrated to match available flood observations. Important points to note are that the national model in this paper simulates flooding in all catchments down to just a few km² whereas the Environment Agency hazard layer may miss some of these. In addition, the local modelling of fluvial flooding often uses 1D hydraulic models rather than the more realistic 1D/2D approach taken here (Bates, 2022). Finally, in coastal areas the local maps are often produced with 'bathtub' GIS mapping rather than true hydrodynamic approaches. 'Bathtub' methods assume that any land under the estimated extreme coastal water level is inundated during an event, even if this is physically impossible in practice due to dynamical effects and lack of hydraulic connectivity. In open coastal plains, 'bathtub' methods may therefore be an extreme over-estimate of the true hazard area (Bates et al., 2005; Vousdoukas et al., 2016). As a result, we should not expect an exact match in all locations, but instead are looking for broad consistency between the two models when the data are examined at scale.

We compare the complete set of local maps to the national layers we compute the following standard metrics:

    a.   Hit Rate: proportion of benchmark data replicated by Fathom model, penalising only underprediction. 0 = none of benchmark captured, 1 = all benchmark captured.

    b.   False Alarm Ratio: ratio of false positives to true positives, penalising only overprediction. 0 = no overprediction, 1 = total overprediction.

    c.   Critical Success Index: overall skill score, accounting for both under and over prediction. 0 = no skill, 1 = perfect skill.

    d.   Error Bias: ratio of over and underprediction errors. 0 = complete underprediction, 0.5 = unbiased, 1 = complete overprediction.

For more details and the equations for each metric see Wing et al. (2017). In Table 1 we give the aggregate performance results for England and then Wales as a whole (top two rows) and for each English region separately, whilst in Figure 2 we compare national and local hazard maps for a variety of inland and coastal locations across the UK. Performance scores for these specific sites are also given in Table 1.





| Scale | HR | FAR | CSI | EB |
|---|---|---|---|---|
| *National* | | | | |
| England | 0.71 | 0.11 | 0.65 | 0.24 |
| Wales | 0.83 | 0.10 | 0.76 | 0.34 |
| *Regional* | | | | |
| North East | 0.75 | 0.25 | 0.60 | 0.50 |
| North West | 0.82 | 0.23 | 0.66 | 0.57 |
| Yorkshire and The Humber | 0.66 | 0.12 | 0.61 | 0.20 |
| East Midlands | 0.73 | 0.09 | 0.68 | 0.22 |
| West Midlands | 0.75 | 0.10 | 0.69 | 0.25 |
| East of England | 0.63 | 0.06 | 0.60 | 0.10 |
| London | 0.86 | 0.26 | 0.66 | 0.68 |
| South East | 0.73 | 0.11 | 0.68 | 0.24 |
| South West | 0.78 | 0.09 | 0.73 | 0.26 |
| *Specific sites in Figure 2* | | | | |
| Tewkesbury | 0.94 | 0.11 | 0.84 | 0.66 |
| The Wash | 0.66 | 0.08 | 0.62 | 0.14 |
| Somerset coast | 0.87 | 0.1 | 0.79 | 0.42 |
| Greater London | 0.9 | 0.26 | 0.68 | 0.76 |

**Table 1: Validation metrics when comparing the 1 in 100-year return period fluvial and 1 in 200-year coastal hazard layer from the model developed in this paper to equivalent government flood maps in England and Wales.**





265



**Figure 2: Comparison of fluvial and coastal hazard layers produced by the national model developed in this paper (left hand panels, labelled 1, in blue) to equivalent government flood hazard maps (right hand panels, labelled 2, in green). Maps are shown for: (a) fluvial flooding surrounding Tewkesbury in Central England at the confluence of the Severn and Warwickshire Avon rivers; (b) predominantly coastal flooding around the area of The Wash tidal embayment in Eastern England; coastal and river flooding in the Somerset Levels, South-West England; and (d) tidal and fluvial flooding in London. Base map data are © OpenStreetMap contributors 2022. Distributed under the Open Data Commons Open Database License (ODbL) v1.0.**

Table 1 and Figure 2 show a coherent match between the two modelled layers with an overall CSI of 0.65 for England and 0.76 for Wales. CSI is a challenging metric as it penalises both over- and underprediction and ignores large areas of non-floodplain that are easy to predict. The metric is also sensitive to the shoreline length to inundated area ratio (Stephens et al., 2014) so that what constitutes a 'good' match varies between sites. CSI values are therefore invariably less than 1 and this results from both model uncertainties (Hocini et al., 2020) and errors in observed data (Hawker et al., 2020; Horritt et al., 2001), both of which can be significant. To put the CSI scores achieved in this paper in context, comparison of modelled flood inundation extent with observations from airborne or satellite sources for individual river reaches typically results in CSI values in the range 0.65-0.9 (Aronica et al., 2002; Horritt & Bates, 2001a, 2002), with the higher value only ever achieved for sites with very high quality input and validation data (Bates et al., 2006; Neal et al., 2009). Similar CSI values to these have also been obtained on the few occasions when separate remote sensing systems have acquired simultaneous images of the same flood (Bates et al., 2006; Biggin and Blyth, 1996; Schumann et al., 2009). Regional validation studies tend to produce slightly lower aggregate CSI values, mostly because regional models, unlike local ones, are never calibrated or optimized to fit the observed data. Example CSI values in regional scale inundation modelling studies to date therefore include 0.36-0.43 in Ward et al. (2017), 0.56-0.67 in Sampson et al. (2015), 0.76 in Wing et al. (2017) and 0.78 in Bates et al. (2021), which also shows the general pattern of improvement over time as regional models have become more sophisticated.

The national model developed in this paper and the local ones developed by the UK's devolved administrations therefore have differences similar to those between local models and observations or between simultaneous observations of the same flood using different sensors. The performance of the UK national model is also in line with that of similar models created for other territories (Bates et al., 2021; Wing et al., 2017). We conclude that our model is a plausible representation of the UK flooding system, with errors in inundation extent likely similar to those in either observations or local models. At the four sites examined in Figure 2 the similarity to government hazard maps is good for the fluvial flooding examples (panels a and d, CSI values of 0.84 and 0.68 respectively) and the coastal and river flooding in Somerset (panel c, CSI=0.79). However, larger and visually obvious discrepancies are shown for the predominantly coastal flooding that occurs around The Wash tidal embayment in Eastern England (panel b, CSI=0.62). The most obvious explanation for the differences at this latter site is that UK Government flood modelling in coastal areas is often undertaken with simple GIS-based 'bathtub' approaches rather than the mass and momentum conserving 2D hydrodynamic methods which are deployed here. Metadata are not available from the Environment Agency in England to show what sort of modelling was conducted for The Wash, but the use of a bathtub scheme is one likely explanation for the differences we observe.

Next, we compared modelled historic water depths to high quality observations of maximum water height for a major flood that occurred in the UK city of Carlisle in 2005. Flooding occurred in the city centre and extensively through surrounding





districts, with approximately 1900 homes inundated. Subsequent to the event, survey teams from the Environment Agency and the University of Bristol mapped wrack and water marks using differential GPS systems (Neal et al., 2009) with a precision of <0.01m in the horizontal and vertical. Wrack marks are defined as trash lines that are assumed to represent maximum flood

extent, whereas water marks are discolouration of vertical surfaces within the flooded area that are thought to represent maximum water elevation. Combining these surveys yielded a set of 263 measurement points spread widely over the Carlisle urban area. These represent one of the most comprehensive model validation data sets for urban flooding currently available. Interpreting wrack and water marks post-event can be difficult, however data quality was evaluated by Fewtrell et al. (2011) who concluded that the mean error across the whole data set was around ±0.1m, whilst likely maximum differences between

individual wrack or water mark measurements and true peak water levels were in the range 0.3-0.5 m.

Parkes and Demeritt (2016) estimate the return period of the 2005 Carlisle flood to have been ~260, years, albeit with high uncertainty (95% confidence interval of 70-4060 years) as is typical for large floods. This observed return period is conveniently close to the 1 in 250-year return period hazard layer simulated by the model, so we therefore compare these predicted maximum water elevations to the observations (see Figure 3).



(a)

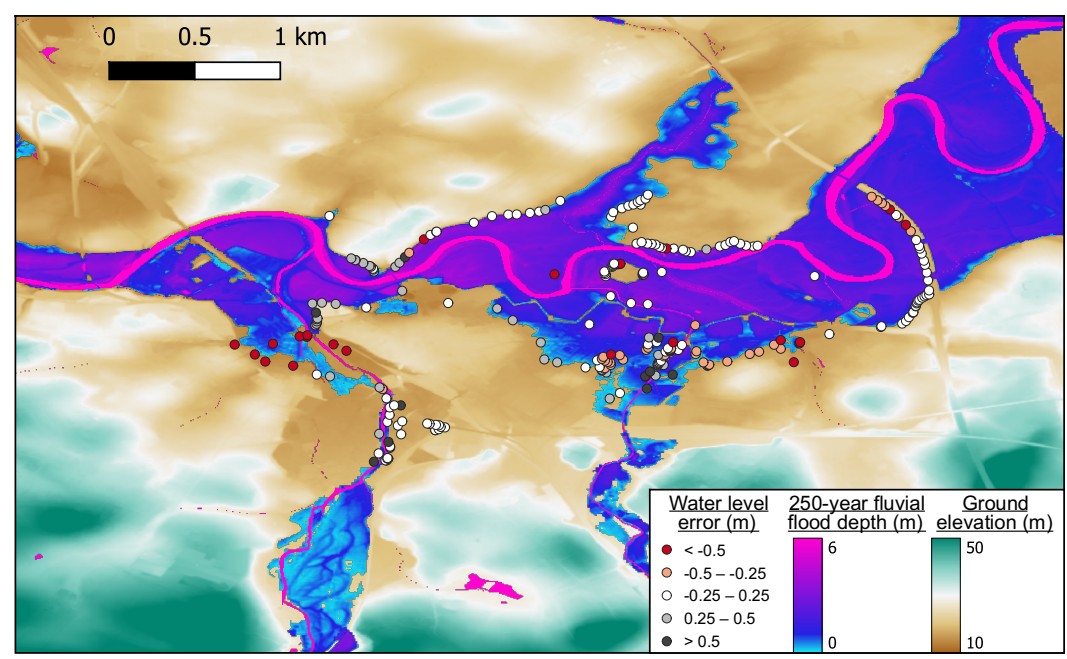

(b)                                                    (c)

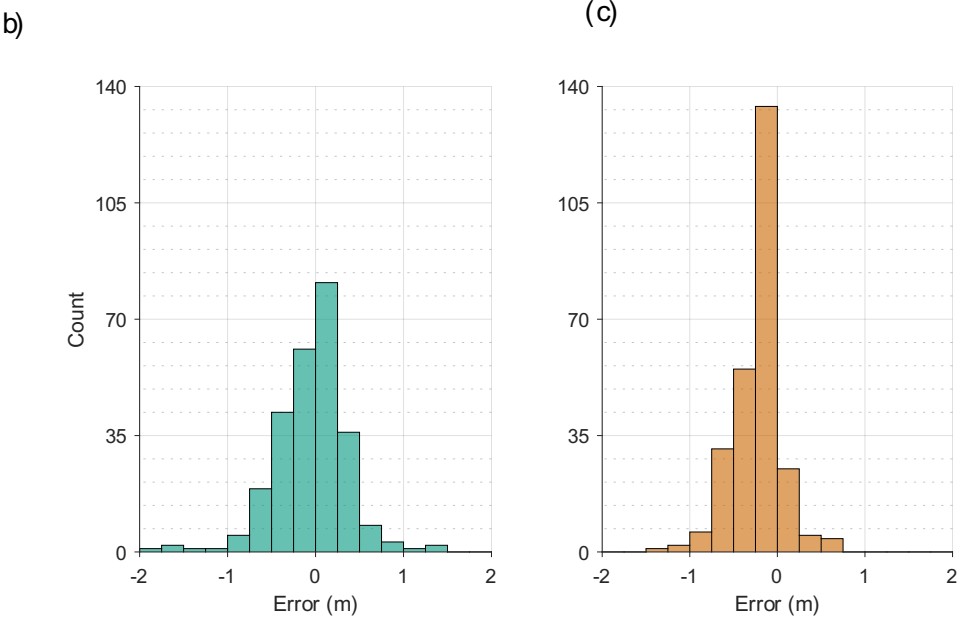


**Figure 3: (a) Model predicted 1 in 250-year water depth and extent for Carlisle showing location of observed maximum water elevations colour coded by error magnitude; (b) frequency histogram of errors for the model reported in this paper; and (c) frequency histogram of errors for a 5m simulation using the 1D/2D hydrodynamic model LISFLOOD-FP built with local flow and bathymetric data and using calibrated friction parameters.**



Simulating maximum water elevations in a dense urban area is not straightforward and the Carlisle simulation presents a difficult test for any hydrodynamic model. Nevertheless, comparison of the national model to observations yielded Root Mean Squared Error of 0.41m, Mean Error of -0.04m and Mean Absolute Error of 0.29m (see Figure 3a and 3b). For context, a 5m spatial resolution LISFLOOD-FP (Bates et al., 2010) model of the Carlisle 2005 event built, following the work of Neal et al. (2009), with observed bathymetry and gauged flows and using calibrated friction parameters gave an RMSE of 0.36m (see

Figure 3c). Both the local and national model errors are greater than observational error, probably as a result of errors in the gauged flow and regional flood frequency analysis procedure respectively. However, in terms of overall RMSE they differ by less than the LiDAR terrain data error at this site (~15cm). The error histograms in Figure 3b and c show larger outliers in the national model which is expected given the coarser spatial resolution of this simulation and the fact that it does not represent in-channel structures such as bridges that the local model includes. Indeed, Figure 3a does seem to show larger errors in the

national model in the vicinity of known structures and future work may need to develop a national database of these obstructions similar to the recently released GROD global product (Yang et al., 2022).

Finally, the Expected Annual Damage produced by the catastrophe model was validated by comparison with the observations of annual insured losses compiled by the Association of British Insurers (ABI) discussed in Section 2.5 and given in Table S1. To convert the ABI's residential-only losses from 1998 to 2018 for the whole UK to combined residential and non-residential

losses in 2020 values for GB only (as produced by the catastrophe model) we broadly follow the approach outlined in Penning-Rowsell (2021). We therefore adjust the ABI data to make allowance for under-insurance, the ABI's incomplete market share and observations of the ratio of residential to non-residential losses from past UK flooding episodes. We can also compare these adjusted EAD values to previous model analyses of UK flood risk undertaken by the Environment Agency in their National Flood Risk Assessment programme (NaFRA, see Sections 2.2 and S1.2) and the third UK Climate Change Risk

Assessment (CCRA3, see Section 2.3 and S1.3). To convert NaFRA and CCRA3 estimates of economic loss to financial values we need to allow for betterment and taxation, and to adjust the NaFRA EAD so that it also represents losses due to pluvial flooding. All data sets are normalised for inflation and increasing Gross Domestic Product to 2020. Further details are given in Section S2. Following these adjustments to bring the data onto a consistent basis we obtain the EAD values reported in Table 2.


| Source | NaFRA | CCRA3 | ABI | This paper |
|---|---|---|---|---|
| Adjusted EAD value (Billions) | £2.246 | £2.154 | £0.714 | £0.730 |

**Table 2: Estimated Annual Damage values for previous modelled analyses (NaFRA, CCRA3), observed insured losses (ABI) and the new model analysis conducted for this paper. These figures represent direct financial losses due to fluvial, pluvial and coastal flooding in 2020 values for residential and non-residential properties in Great Britain.**

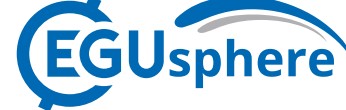

The small deviation between our modelled EAD and the ABI data is pleasing, and likely within the range of observational error (resulting from approximations during loss adjustment, errors in reporting etc). However, the 20 years of ABI historical observations represent just one realisation of the insured losses that could potentially occur during this period. The catastrophe model developed here simulates a 10,000-year synthetic catalogue of flooding and therefore includes very low probability, high loss events that may not be present in the ABI's historical record simply due to chance. To get a sense of likely uncertainty

in the ABI data as a result of the under-sampling of very extreme events, we randomly selected 10,000 periods of 20 years from the catastrophe model and calculated the EAD for each of these. The frequency histogram for these random 20-year samples is reported in Figure 4, along with vertical lines representing the ABI, NaFRA and CCRA3 Expected Annual Damages.

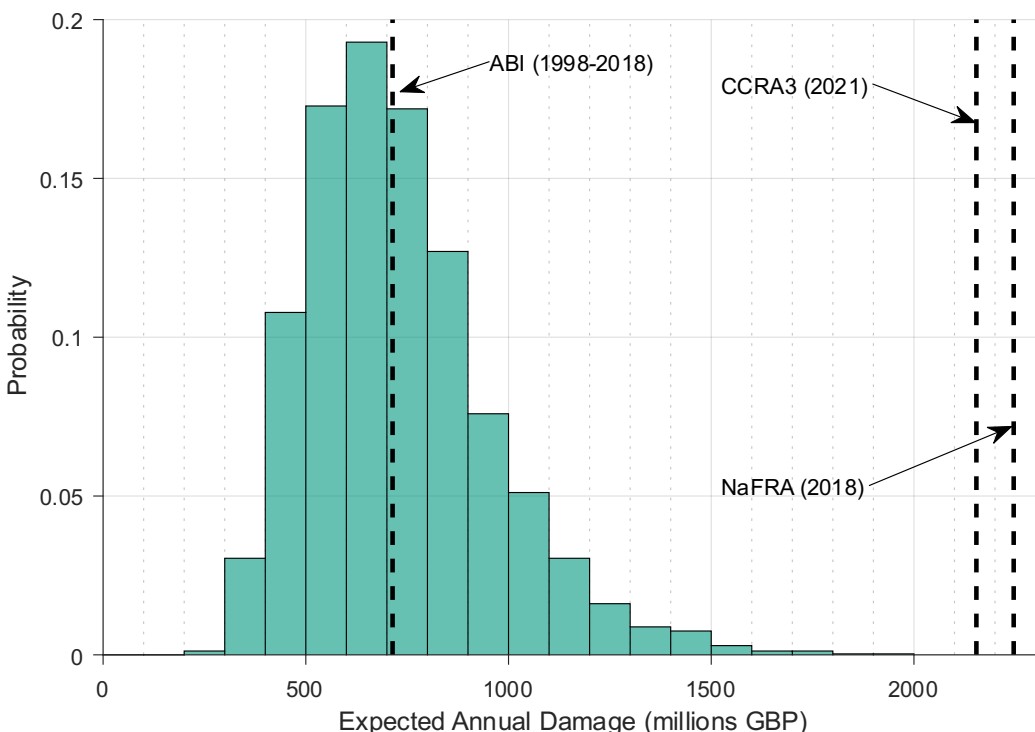

**Figure 4: Simulated EADs from 10,000 random samples of 20-year time periods for the model developed in this paper for 2020**
**conditions compared to adjusted values from previous model estimates from NaFRA, CCRA3 and observations from the ABI.**

    Figure 4 makes clear the likely large impact on EAD of random sampling of very extreme floods during any finite period of historical data. Nevertheless, the observed ABI data sit squarely within our bootstrapped loss distribution. Expected Annual Damages from previous UK model analyses (NaFRA, CCRA3) are three times larger than ABI observed losses (as previously noted by Penning-Rowsell, 2021) and lie well outside the distribution of 20-year samples of loss from the catastrophe model

reported here. Based on our distribution of losses, one would expect values equivalent to NaFRA and CCRA3 Expected Annual Damages to occur every ~15 years and not annually. Whilst the catastrophe model developed here is not truth, Figure 4 does indicate that the differences between NaFRA/CCRA3 EADs and the ABI observations are very unlikely to be the result





of extreme losses included in the NaFRA analysis and missing from the historical record. Instead, it suggests that the NaFRA and CCRA method may have significant errors.

**4.2 Risk projections and climate change**

Based on the hazard and risk validation evidence presented above, the catastrophe model developed here appears to be a reasonable representation of UK flood patterns and losses. To examine how climate change will impact flooding in Great Britain we calculate loss-exceedance curves in 2020 monetary values from the catastrophe model for specific global warming levels above pre-industrial (1850-1900). The loss-exceedance curve shows the probability that a particular total annual loss

will be exceeded in any given year. These are shown in Figure 5, with values for the average annual and 1% annual probability (1 in 100 year) loss given in Table 3.

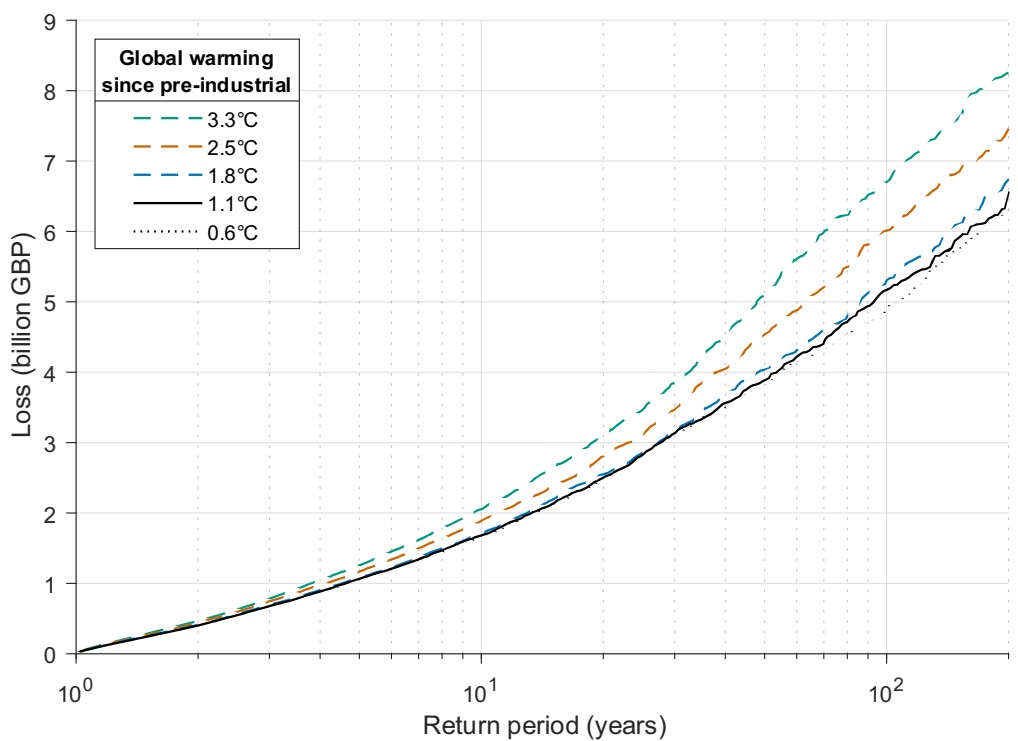

**Figure 5: GB loss-exceedance curves for different specific global warming levels since the pre-industrial (1850-1900).**





| Warming level (°C) | EAD (£Bn) | % change | 1 in 100 year loss (£Bn) | % change |
|---|---|---|---|---|
| 0.6 | 0.73 | - | 4.89 | - |
| 1.1 | 0.74 | 1.5 | 5.17 | 5.7 |
| 1.8 | 0.76 | 3.7 | 5.30 | 8.4 |
| 2.5 | 0.83 | 13.2 | 6.01 | 22.9 |
| 3.3 | 0.90 | 23.0 | 6.70 | 37.0 |


**Table 3: GB expected annual and 1% annual probability flood losses in £Bn in 2020 values for different specific global warming levels since the pre-industrial (1850-1900).**

A specific global warming level of 1.1°C approximately represents the present day (2020 in our case), whilst a threshold of 0.6°C was crossed sometime around 1995 and is therefore equivalent to the historical run of our model. 1.8°C represents the
likely maximum warming under the Paris Agreement target if the 2030 emission reduction pledges made at COP26 are implemented in full and, in addition, countries' longer term 'net zero' ambitions are realised by the mid-21st century (Hausfather and Moore, 2022; Meinshausen et al., 2022). A specific global warming level of 2.5°C above pre-industrial is approximately what is projected if only the COP26 2030 emission reduction targets are achieved and then $CO_2$ reductions stall, whilst 3.3°C gives an indication of what is likely to happen if the Paris Agreement and 'net zero' targets are missed and climate
sensitivity turns out to be towards the upper end of the current plausible range. In all these calculations we assume 2020 population and assets: future work will look at the balance between socio-economic changes and climate change on future flooding. Where this balance has been examined in other territories (Swain et al., 2020; Wing et al., 2018, 2022) population change is typically shown to be a significantly larger driver of future risk than changes in precipitation and temperature.

Figure 5 shows that, according to our model, changes in flood losses over Great Britain due to climate change alone between
recent historical average conditions and the present day have so far been minor, with differences only really emerging for annual loss return periods greater than 70 years. The increase in EAD between these warming levels is only 1.5%, whilst the 1% annual probability (1 in 100-year return period) loss with 1.1°C of warming is £5.17Bn, compared to £4.89Bn with 0.6°C of warming, a~ 6% increase. This is consistent with the already-observed (but modest) changes in extreme rainfall resulting from anthropogenic climate change (Kendon et al., 2021). Figure 5 also shows that Great Britain will only be able to avoid
major increases in flood risk due to climate change if all countries' current COP26 and 'net zero' emission reduction pledges are met in full and warming above pre-industrial is limited to 1.8°C. In this policy scenario, the UK 1% annual probability flood loss of £5.3Bn represents only a ~8% increase above recent historical conditions, whilst AAL increases by only ~4%. However, if the 'net zero' targets are missed then much larger increases in AALs are possible: ~13% for 2.5°C and ~23% for 3.3°C. 1% annual probability losses rise by even greater amounts to £6Bn for 2.5°C and £6.7Bn for 3.3°C. Respectively,



these represent 23% and 37% increases over recent historical (0.6°C of warming) conditions and are significant in terms of both the required annual flood defence spending to adapt to these risks and the capital provision against flood losses that financial markets will need to make.

The curves in Figure 5 represent UK aggregate losses, however this conceals important changes in the geography of risk (Figure 6). Figure 6 shows absolute Expected Annual Damage aggregated to 10km hexagons across Great Britain for historical

average conditions (0.6°C of warming, Figure 6a, left hand panel) and the percentage change in EAD assuming that the current COP26 2030 commitments and 'net zero' pledges are implemented by all countries on time and in full (1.8°C of warming, Figure 6b, right hand panel). Historical EADs are, unsurprisingly, largest for 10km spatial units containing major population centres (London, South Wales, the cities of the Liverpool-Manchester area and the Glasgow-Edinburgh region of the Scottish Central belt). Moreover, whilst changes in national aggregate EAD above recent historical values for Great Britain under

1.8°C of warming are modest, Figure 6 shows significant spatial patterns that include areas of significant increase (>25%) as well as areas with small decreases. Significant increases occur over South-east England, South and West Wales and North-West England. More modest increases occur over much of Central and Western Scotland whilst small decreases in flood losses occur over South-West, Central and North-East England. These patterns result from the complex interplay of changing flood drivers (see Figure S2) with asset exposure. Broadly, extreme rainfall and sea level increase almost everywhere (albeit

by a spatially variable amount that varies over time) and these drive increased pluvial and coastal flooding. However, warmer future temperatures mean that increased rainfalls can, at times, fall on drier soils and result in less catchment runoff. Fluvial flood hazards therefore decrease across large areas of Eastern, Central and South-Western regions and this can counter-balance increases in other flood drivers, resulting in net risk reductions at 10km scale. Resolving and analysing more detail than this would likely represent over-confidence in model skill, as the fine-scale spatial patterns would likely change somewhat with

different climate projections and hydrological modelling. Nevertheless, we expect the broad regional patterns in our data to hold and be useful in developing policy. Lastly, this complex pattern of both increases and decreases in rainfall and river flow is very different to current UK climate change allowances for rainfall and river flow (Kay et al., 2021) which are almost uniformly positive for all future time periods (see Section S3.1.2 for further details). A further observation from Figure 6 is that many areas of rising risk are locations where the risk is already high (e.g., London and South Wales).



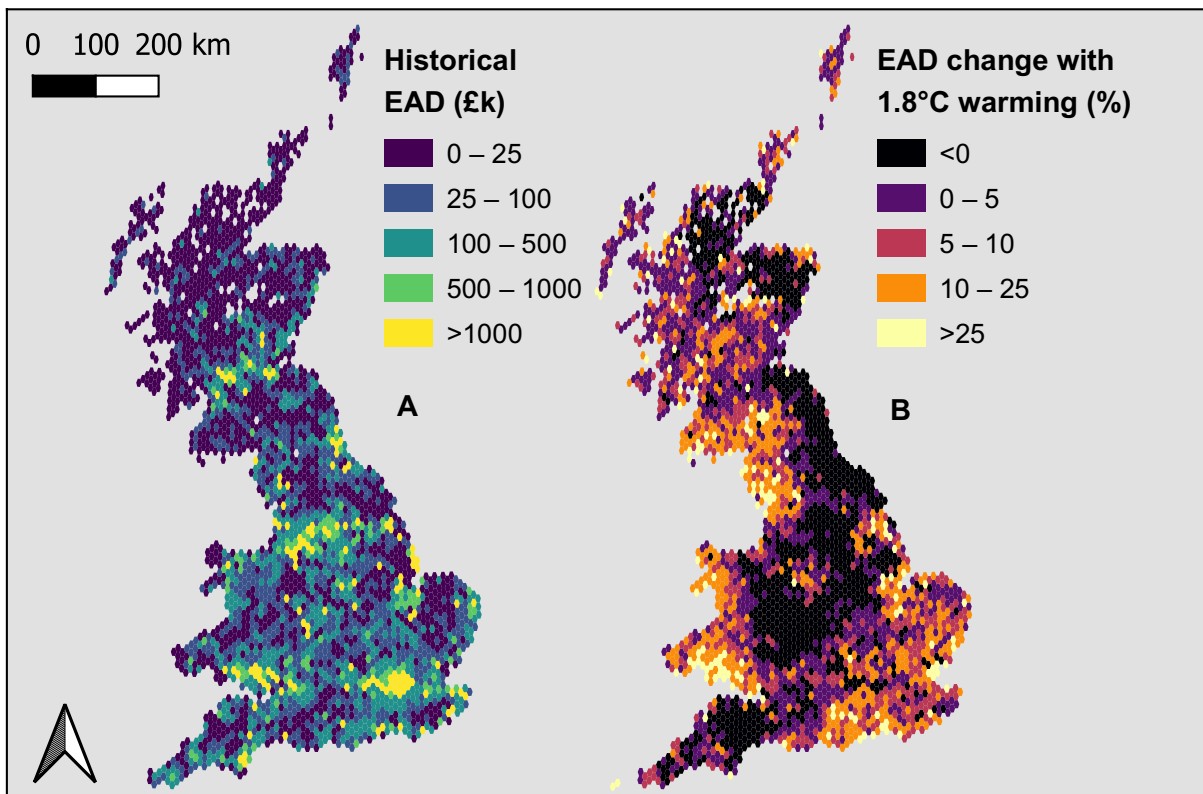


**Figure 6: Spatial distribution of Expected Annual Damage due to flooding for historical conditions (0.6°C of warming) and percentage change between this and a warming level of 1.8°C which approximately represents a world in which meet the Paris Agreement targets.**

## 5. Conclusions

Current assessments of flood hazard and risk in the UK lack transparency, are insufficiently validated and, with very few exceptions, are not exposed to independent peer-review. Whilst the public availability of the data sets is impressive by international standards, the methods used in their creation are clouded in secrecy. No details of the methodologies used by Wales and Scotland are in the public domain, and only limited information is available for England and Northern Ireland. Calls for proper peer-review of UK national flood risk assessments have been made before (Penning-Rowsell, 2014), but have

effectively been ignored. The methods are therefore not repeatable by others.

This situation slows down and hampers model review and improvement cycles whilst restricting the number of researchers that can contribute to the effort, thereby creating significant barriers to progress. To give an example of this in operation, the NaFRA methodology used in England was developed by Hall et al. (2003) with a new major version in 2008 and no change since then (Penning-Rowsell, 2021). A wholesale revision of the methodology (NaFRA2) is currently underway and is

scheduled to come into operation in 2024 or 2025. Major model updates are thus ~decadal and following the initial peer-reviewed work in 2003, no details of methodological revisions have appeared in the public domain. By contrast, peer-reviewed





large scale inundation modelling methods have seen rapid advances from the first national scale models in data rich countries in 2004 (Bradbrook et al., 2005), to simulation of globally significant data poor basins in 2007 (Wilson et al., 2007), simple global methods in 2013 (Ward et al., 2013) and full 2D hydrodynamic global models at <100m resolution by 2015 (Sampson

et al., 2015). This has required technological developments in the use of high-performance computing, modelling methods, data sets and machine learning algorithms that are contained in a now large body of literature (e.g., Addor et al., 2020; Alfieri et al., 2016; Allen & Pavelsky, 2018; Hawker et al., 2022; Knox et al., 2022; Morales-Hernández et al., 2020; Neal et al., 2021; Yamazaki et al., 2017; Zhao et al., 2021 to cite just a few). Numerous significant advances now occur every year and the rate is accelerating. It is thus abundantly clear that peer-review accelerates the model development cycle, allows rapid diffusion

of best practice and enables contributions from an increasing pool of new researchers who can bring novel ideas. Flood risk management in the UK does not currently take advantage of this engine for progress.

This paper attempts to kick start this process for the UK by demonstrating a first climate-conditioned catastrophe risk model for UK flooding which shows skill at simulating both hazard and risk. A model-to-model comparison with official 1 in 100-year return period fluvial and 1 in 200-year coastal hazard maps across the whole of England and Wales gave an overall Critical

Success Index (CSI) value of 0.65 for England and 0.76 for Wales. CSI is a challenging metric to maximize as it penalises both under and over-prediction, and ignores easy to predict dry areas, but this value is similar to that obtained in comparisons of local hydrodynamic models to remote sensing observations of flooding (Aronica et al., 2002; Horritt & Bates, 2001, 2002). CSI values over the UK are slightly lower than those obtained in a similar study over the US (Bates et al., 2021), in part because the US has more big rivers which are, in general, easier to model. CSI values for the UK are also influenced by the

different methods employed in coastal areas by the local and national models i.e., 'bathtub' GIS models used in local studies and the hydrodynamic approach used here at national scale which is likely to be more accurate. In fluvial areas the similarity between the local and national models is generally better (see Figure 2, panels a and d), and comparison of maximum water levels predicted by the national model for the 2005 Carlisle flood gave RMSE of 0.41m compared to 0.36m for a high-resolution local model that was built and calibrated with site specific data. Most importantly, the national model was also able

to provide a good match (i.e., one that is within likely error) to observed annual flood losses from the Association of British Insurers (ABI).

Our model analysis of course comes with a number of caveats. Driving the analysis with different climate models would change the detail of local predictions, however there is agreement on the broad patterns of UK climate change and the 12km Regional Climate Model used in UKCP18 is the current official estimate of UK future climate. Hydrological modelling

contains significant uncertainty arising from the boundary forcing, input data, model parameters and calibration (Beven, 2006; Coxon et al., 2019), and hydrodynamic models are predominantly sensitive to DEM and forcing errors as well as the quality of nationally available flood defence information. The latter is likely a key limiting factor for any large-scale flood inundation analysis, and whilst some work-arounds are possible (e.g., Wing et al., 2019) these are by no means perfect. We also use a single change factor for all event return periods which may be an oversimplification. More sophisticated work could use a

multi-model ensemble of climate model simulations (e.g., Cloke et al., 2013), run multiple simulations to account for



uncertainty (e.g., Keef, et al., 2012), use better flood defence information and higher model resolution over urban areas (Fewtrell et al., 2008) and take into account the probability of defence failure (Shustikova et al., 2020). Nevertheless, the model simulations shown here do have skill and represent a significant advance on previous work such that there can be confidence in the broad conclusions that we draw.

Expected Annual Damage (EAD) due to flooding in official UK data gives values that are ~3x higher than the observed ABI values, as Penning-Rowsell (2021) has previously observed. Moreover, these government estimates are used more or less directly in the UK's Climate Change Risk Assessment (CCRA) process, so this analysis also inherits these biases. Whilst the ABI data need careful handling and adjustment because of the way they have been collected, these differences are stark. Nevertheless, the ABI data provide only a 20-year snapshot of possible UK flood losses whilst official model data include

very large events (up to 1 in 1000-year return period) which are physically possible but may not be present in the historic record. Such events may be large contributors to EAD despite their low probability. However, the analysis developed in this paper shows that this is very unlikely to account for the difference between observed flood losses and official estimates as the latter lie well outside our modelled distribution of 20-year losses which is plausibly centred on the observations. This distribution (which is a surrogate for natural climate variability) means that the 5-95% range for EAD is 0.43-1.14 £Bn in our

model, so the official estimate of £2.4Bn seems implausibly high.

Our modelling shows that, even with a more sensible loss distribution than official UK government estimates, the COP26 2030 pledges on decarbonization are not on their own sufficient to restrict increases in UK flood risk to <10% for either EAD and/or 1% annual probability flood loss. Instead, International 'net zero' commitments will need to be implemented in full else rises in 1% annual probability loss of 23% or even 37% are possible depending on the pathway that societies take. Adaption to

these greater increases is still likely possible for the UK, but at much increased cost. Whilst the most optimistic climate scenarios see only modest increases in losses at national scales, this conceals a dramatically changing geography of risk. These spatial variations are significant and create both winners and losers, with some places seeing ~25% increases in the 1% annual probability loss even if 'net zero' pledges are implemented in full. It is therefore strongly in the UK's interest to exercise leadership in carbon emission reductions, both by example and as part of global diplomatic efforts.

It is also clear that the UK is not well adapted to the flood risks it currently faces, let alone any further increases in risk due climate change. Current Expected Annual Damages of ~£700M are a drain on the economy, but more importantly this represents a very considerable sum of misery for those who are affected (e.g., French et al., 2019). Most places in the UK that will be at risk of flooding in the future are already at risk now. It follows that the best thing we can do to prepare for the impact of climate change is to strengthen flood management in currently at-risk areas, and this will have immediate economic and

social benefits as well.

In summary, we have presented a plausible and sober assessment of current and future UK flood risk. The analysis contains a greater level of detail and nuance compared to previous work and represents our current best understanding of the UK's changing flood risk landscape. Whilst we should be cautious of over-interpreting the fine scale spatial detail of the predictions, we expect that the national scale results and broad regional patterns can be used in framing policy. The complexity of the



climate-driven change we find in UK flood risk is likely to ring universally true in other parts of the world and should cause us to question simplistic flood risk projections and policy responses.

## Acknowledgements

Paul Bates was supported by a Royal Society Wolfson Research Merit Award. The work in this paper was in part supported by UK Natural Environment Research Council grant NE/S006079/1. Jeffrey Neal's involvement in the work was funded by UK Natural Environment Research Council grant NE/S015795/1. Funding for the work in this paper was also provided by Fathom (www.fathom.global).

## Data availability

A version of the LISFLOOD-FP model similar to that used in this work is available at https://zenodo.org/record/4073011#.YuJ0W3bMKUl. For academic access to the data sets created in this paper please contact o.wing@fathom.global.

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
