# Peer review of "A climate-conditioned catastrophe risk model for UK flooding"

_EGUsphere, 2022_

## Author Response (AR1)

Professor PAUL BATES
B.Sc., Ph.D.
Professor of Hydrology

**SCHOOL OF GEOGRAPHICAL SCIENCES**
University of Bristol
University Road
Bristol
BS8 1SS
UK
Tel: +44 (0)117 928 9954
Fax: +44 (0)117 928 7878
E-mail: geog-office@bristol.ac.uk
www.ggy.bris.ac.uk

| *From*: | Professor Paul Bates |
| *Tel:* | +44-117-928-9108 |
| *E-mail:* | Paul.Bates@Bristol.ac.uk |

Professor Bruce Malamud
Editor
Natural Hazards and Earth System Science

**Re. Author response for Bates et al.**

Dear Bruce,

I'm really delighted that the work was well received and I'm very grateful for the very kind words from both reviewers.

We have revised the manuscript to take full account of the review comments with the following modifications as per our posted responses in the open discussion. In the following we show the referee comment in black, the posted response in blue and the specific changes made to the manuscript in red. All line numbers given in the revision letter refer to the new 'tracked changes' version of the manuscript. I have also included a table at the end showing how we have dealt with your editorial comments.

In addition, we have made a number of other minor typographic changes to improve the readability and accessibility of the manuscript.

Yours sincerely,

Paul Bates

**Referee RC1**

This paper summarizes the impressive work done by Paul Bates and colleagues in flood risk modelling. While the details are left to other more technical papers, this work concentrates on the big picture and answers to the call for peer-reviewable flood risk assessments (in the UK but I would say also elsewhere). Because of this, I believe that the paper fits is appropriate for NHESS. I only have a couple of minor suggestions (and some more detailed comments below):

> We are very grateful to the reviewer for their supportive comments and the suggestions for improvement.

- Despite the title and abstract focus on climate scenarios, in the paper, a lot of emphasis is given to the transparency and consistency of the hazard and risk maps obtained through this modelling chain as opposed to the "opaque" official maps. The climate change analysis, instead, is not fully developed. The assessment of uncertainty with an envelope of climate models (and for different emission scenarios) would be a standard requirement (and the estimation of additional uncertainties of the hydrologic and hydraulic models would be even better). My suggestion would be to rephrase title and abstract to reflect the weight given in the paper to the product transparency (which has the appealing side effect of allowing coherent change analyses, for which this paper shows an example).

> The reviewer makes a good point regarding the title and abstract and suggests further highlighting how this work address the lack of transparency in 'official' flood maps. This is a problem not only in the UK, but also elsewhere to the best of our knowledge. In a revised version of the paper we will make changes as suggested to address this point and better emphasise this aspect of our contribution.
> We have modified the opening sentence of the abstract (line 7) to better make this point.
>
> We also agree that future work should more fully develop the climate change analysis and look at projecting flood risk using ensembles of climate models and different emissions scenarios. This would be a substantial task and one that would need to build on the work presented here. As the current paper is already 23,000 words long (main text plus supplementary information) we think this would need to be as a separate contribution. We will however modify our paper to acknowledge the limitations of the basic climate change assessment undertaken in this proof-of-concept work and discuss what a robust assessment of climate uncertainty might look like.
> We have added the sentence "However, subsequent work should extend this 'proof-of-concept' to consider ensembles of climate models and a wider range of emission trajectories" at line 219-20.
>
> In addition, we do already state on line 477 of the conclusions that "Driving the analysis with different climate models would change the detail of local predictions".

- The paper is a manifest for the consistent flood risk modelling at the national scale. It is a step toward the construction of the digital twin (of UK in this case). Is there any room for local knowledge in this game? I would expect, and the Authors acknowledge it, that locally tailored models may be more accurate than the national one. This is partly due to the calibration with local data but also to the better specification of boundary conditions, including hot-spots, that is possible because of local knowledge. Is there a way to incorporate local knowledge in a "consistent" large scale modelling effort maintaining its consistency? It would be nice to have a couple of lines discussing this point in the concluding section.

The comment regarding the role of local knowledge in national scale models is also extremely pertinent. Whilst, national scale models need to be built from available and standardized data sets, there is a need to incorporate local knowledge in a consistent and traceable way that does not lead to local over-fitting for locations where validation data exists. Local over-fitting can give validation studies the appearance of rigour but may mean that this apparent level of skill cannot be generalised to other places.

Instead, we need to find ways to: (i) recover and assemble local data (e.g. on river bathymetry, flood defences and validation data) into consistent national databases and (ii) replicate the decision making of skilled local modellers in automated frameworks. The goal should be to create national models with local knowledge and skill, but, as we have seen in the US, doing this whole process manually is not a scalable solution. For example, the Federal Emergency Management Agency's national flood mapping program is based on a patchwork of local models however the total cost from its inception in 1969 to 2020 was $10.6 billion, while covering only 33% of the rivers and streams in the country (Association of State Floodplain Managers, 2020).

In the revised paper we will add text to the conclusion to discuss this important issue and thank the referee for drawing our attention to this point.
To address this point, we have added the following text to the conclusions at line 487:

"We also need to find better ways to recover and assemble local and ad hoc data (e.g., on river bathymetry, flood defences and validation data) into consistent national databases and develop algorithms to replicate the decision making of skilled local modellers in automated ways. Our ultimate goal should be to create national models which have equivalent performance to local approaches."

Detailed comments:

Line 88: Bloeschl et al. (2019, https://doi.org/10.1038/s41586-019-1495-6) show a significant increase of river flood magnitudes over the UK, specially the northern part, which is one of the clearest hotspots in Europe for that matter.

Thanks. This is a useful reference which we will add to the paper.
Added at lines 90 and 406.

Line 212: under the RCP8.5 scenario only? Are therefore the different worming levels correspondent to different future times?

The UKCP 12km regional model simulations we use for the climate projections represent 20-year time slices centred on 2030, 2050 and 2070 under RCP8.5 only. These are the 'official' climate projections produced by the UK Met Office and therefore an obvious choice and starting point for this work. We interrogate these simulations to find the points when particular specific global warming levels are crossed and then present the loss results based on the changed climate to this date. The different warming levels do therefore represent different future times, but an advantage is that this approach gives a degree of scenario-independence. Whilst the RCP8.5 trajectory is increasingly considered unlikely we only use this scenario to extract results at specific warming levels so are making no judgements about its probability.
To address this point, we have added the following text at tine 237-240.

"The different warming levels therefore represent different future times, but an advantage to this approach is that it gives a degree of scenario-independence. Whilst the RCP8.5 trajectory is increasingly considered unlikely we only use this scenario to extract results at specific warming levels so are making no judgements about its probability."

It might also be useful to note that, at least until mid-century, the differences over the UK amongst the different emissions scenarios are relatively small. Because we consider near-future projections of flood risk, the impact of climate scenario choice is therefore limited. We will add further text to make these points clear.
To address this point, we have added the following text at line 217-219:

"Because we consider near-future projections of flood risk, the impact of climate scenario choice is somewhat limited because, at least until mid-century, the differences over the UK amongst the different emissions pathways are relatively small."

Line 216: uncertainty in hydrological modelling is accounted for. How? What are the regionalised "results"? How regionalised?

We simply relate change factors to catchment physical characteristics in different UK regions to extrapolate the set of hydrological model outputs to basins that we have not explicitly modelled. We will add further text to make this point clear.
This sentence has now been modified to read:

"Parameter uncertainty is accounted for in the hydrological simulations using an ensemble approach and the results are regionalised based on catchment physical characteristics to give full national coverage."

Line 220: for how many years are the stochastically generated events simulated over the UK? How many events per year are generated on average? (PS. at page 16 I see it is 10000 years)

The event rate is determined from an empirical distribution fitted to the annual event counts in the historic gauge data. For each year of the 10,000-year simulation, the number of events to be generated was sampled from this distribution. This resulted in ~343,000 events, ~170,000 of which have a >1 in 5-year magnitude event in at least one catchment (so ~17 per year). This is already detailed in the Supplementary Information on lines 432-439.
No change required

Table 1: it would be very informative to also stratify the results by river flooding, pluvial flooding, and coastal flooding.

This would indeed be nice to do, but the data sets we summarize rarely report the information in this way so this unfortunately cannot be done. We can however split our model results by flood hazard type and will add this to the revised manuscript.
Now added at line 401-3.

Line 267: labelled 2, in "red"?

Good spot, this is a mistake and will be corrected.
Now corrected.

Table 2: for "ABI" and "This paper" one could report also other statistics for the annual damages (not only the mean but, for example, the 25% and 75% quantiles, or more) which would show whether the distribution of "observed" annual damages is captured by the model, and that "This paper" is much more informative than NaFRA and CCRA3.

This is a good idea; we will add this.
This has now been added as a shaded area on Figure 4.

Line 479: for past changes, see e.g., Bertola et al. (2020, https://doi.org/10.5194/hess-24-1805-2020).

Thanks for drawing our attention to this very useful reference. We will add this to the paper to support the point made here.
Reference now added.

References
Association of State Floodplain Managers: Flood Mapping for the Nation: A Cost Analysis for Completing and Maintaining the Nation's NFIP Flood Map Inventory, Madison, WI, 2020.

**Referee RC2**

The paper presents a spatially consistent and transparent approach to model flood risk across the UK. The focus of the paper is on comparing the outcomes of a new coupled hydrodynamic – catastrophe model for fluvial, pluvial and coastal flooding with existing approaches, which are in the public domain but insufficiently documented. The authors show that existing approaches used for national flood and climate change risk assessments are likely overestimating the expected annual losses from flooding, due to a number of simplifications, such as when estimating the inundation from coastal flooding.

> We are very grateful to the reviewer for their supportive comments and the suggestions for improvement.

The paper is an important contribution to flood risk modelling in the UK and the resulting flood hazard and risk maps are the first high skill alternative to the official flood maps provided by UK government agencies and should therefore be published in NHESS. To walk the talk, I would like to encourage the authors to make the flood maps for different return periods and climate scenarios available for the academic community under an open-source license for non-commercial use.

> First of all, we'd like to reassure the reviewer that we are indeed "walking the talk" by making the data fully available for non-commercial research use under a standard academic licence.  This is already mentioned in the "Data availability" section on lines 523-525 of the main text but we will see if this could be made clearer.
> We checked the data availability statement, and this does already make clear how academics can obtain access to the results.  We also signpost the data availability statement at the end of the introduction in the existing manuscript.  This is very clear and should be sufficient, so no further change is required.

General comments

The paper addresses the really important issue of a lack of alternatives to the official flood hazard and risk maps in the UK, which are spatially inconsistent and not well documented. The paper is well written, clearly structured and critically reflects on several caveats and limitations of the described approach. I have two main points of criticism, which have already been partly addressed by the authors but could be made clearer.

My first point is in regard to the validation of the estimated EAD from the model against insurance claims data from the ABI. The ABI data must be seen as the lower end of any damage estimation due to a number of reasons of which many are mentioned in the manuscript (e.g. data only covers insured residential properties, data on commercial flood damage not included etc.). I agree with the authors assessment that both NaFRA and CCRA3 likely overestimates the EAD, but I would argue that the author's approach on the other hand is very likely an underestimation of the EAD, which should be discussed in more detail in the manuscript.

> In terms of the ABI data and whether this is an under-estimate or not, it is worth noting that the ABI data have already been substantially adjusted to deal with many of their limitations. For this we follow the approach given in Penning-Rowsell (2021).  This method corrects the data (as far as can reasonably be accomplished at present) for inflation, territorial basis, betterment, taxation, missing pluvial flood losses, underinsurance, ABI market share, missing non-residential losses and changing GDP over time.  There is a long section on this in the SI on lines 170-225 and the approach is summarised in the main text on lines 153-163 and 332-337.  Of course, this is not to claim that the corrected ABI data are therefore 'truth' or that

the correction factors determined by Penning-Rowsell are exact, but it does mean that it is not at all clear that the corrected ABI Expected Annual Damage value is an under-estimate. Post-correction, the ABI EAD is just as likely to be too high as too low. The ABI data will of course have error and we do already note in the conclusions on line 488 that "the ABI data need careful handling and adjustment because of the way they have been collected". However, the referee is correct that we could have said more about their likely uncertainty and will add this discussion to a revised version of the paper.

The text regarding this point on lines 164-6 has been modified to read:

"Despite these caveats, following correction for obvious biases the ABI data do provide a set of realistic observed annual flood losses to compare to modelled estimates. Whilst the ABI data are not 'truth', they do represent our current best empirical data on recent UK flood losses."

Underlined words represent the added text.

My second point is in regard to the climate scenarios. While the loss exceedance curves in Figure 5 and the EAD in Table 3 for different warming scenarios are scientifically interesting, I wonder what we can learn from a scenario that is above current warming levels but with current levels of exposure and vulnerability as we know that such a risk scenario is highly unlikely to occur (it would mean that we stop all human activity in the UK until the 2030s to make for example the 1.8°C scenario presented by the authors a credible one). In my opinion the spatial analysis shown in Figure 6 is more meaningful as it allows to see spatial changes in the hazard under climate change (although I would think it makes more sense to interpret those changes in qualitative terms).

We agree that including socio-economic as well as climate scenarios would be interesting, however understanding changes in risk due to climate alone is extremely useful in its own right. Moreover, only by controlling for socio-economic change can the impact of particular climate emission policy responses be clearly identified. Demarcating the impact on flood risk of the COP26 commitments and 'net zero' ambitions is major outcome of the paper and should have wide impact. The reviewer is however correct that the next step is to look at the interplay between climate and development in modulating future risk, although this is not trivial because of the granularity of socio-economic projections that are required. The IPCC Shared Socio-economic Pathways (SSPs) are at country level and downscaling of these to 1km (i.e., still much coarser that the ~20-25m resolution inundation model) has only just been completed for the UK (see https://uk-scape.ceh.ac.uk/our-science/projects/SPEED/shared-socioeconomic-pathways). These downscaled SSP data will need careful evaluation prior to their use in a flood risk study and some careful methodological development will be needed to bridge the remaining resolution gap. This will be a substantial task and one that realistically will need to be described in a stand-alone paper. Including socio-economic projections in the present (rather overlong) manuscript is probably too much. Instead, we do already include statements about the likely impact of including socio-economic change on lines 390-393 and indicate that this should be looked at in future work.

We feel this point is already sufficiently well addressed by the existing text where we state:

"In all these calculations we assume 2020 population and assets: future work will look at the balance between socio-economic changes and climate change on future flooding. Where this balance has been examined in other territories (Swain et al., 2020; Wing et al., 2018, 2022) population change is typically shown to be a significantly larger driver of future risk than changes in precipitation and temperature."

No further change therefore required.

I would not expect the authors to significantly change their results, but provide a bit more context how they would like readers to interpret their results.

Specific comments:

P7 L194ff: One main advantage of the local modelling approach used by the Environment Agency is that they have a good understanding of local flood defences and other protection infrastructure. Can you say something about how your approach compares to that? I have not checked Wing et al. 2019, but in case you have any information on the accuracy of your approach compared to data on local spatial flood defences that would be great.

Primarily, we use the exactly same government flood defence database as the UK environmental agencies, namely AIMS (https://www.data.gov.uk/dataset/cc76738e-fc17-49f9-a216-977c61858dda/aims-spatial-flood-defences-inc-standardised-attributes). A reference to this is already included in the bibliography. Most flood defences in our model are based on this 'official' view and therefore the majority should be exactly the same between local and national studies. It is also worth noting that local models are only employed in the UK to produce estimates of flood hazard, and these hazard maps are not currently used in the production of national risk estimates. Instead, flood risk is determined separately using large scale simplified inundation models built using national data sets including AIMS for the flood defences (e.g., the NaFRA methodology in England). Official national scale risk estimates thus do not benefit from local knowledge either (at least as far as we can tell from the limited information about these methods that is in the public domain).

The referee is correct however that that EA, SEPA, NRW and DfI local flood hazard modelling studies may possibly supplement the AIMS data with knowledge that is not systematically recorded in an open-source form. Large scale studies, as conducted here, need to work with available published data and their results may diverge from local modelling where this such information has a significant impact. To address some of these limitations we use the method of Wing et al. (2019) to automatically identify flood defences in high resolution terrain data and apply this everywhere such data exists. The Wing et al (2019) paper showed that this method could added important information to official flood defence records and for a test reach of the River Po led to improved model predictions. Importantly, the method can identify structures which impact flood propagation on floodplains, such as causewayed roads and railway embankments, which are not officially classified as flood defences. It is difficult to generalize the River Po findings, but in general we would expect this automatic detection approach to miss some flood defences that local knowledge would pick up, but at the same time it may identify relevant terrain features that might otherwise be overlooked. NaFRA does not include a similar methodology to supplement AIMS flood defence information (as far as we can tell).

We will add some further comments to the paper to discuss this.
We have extensively modified the text around lines 200-208 to address this point.

P8 L199: How where the 10 different return periods selected? Olsen et al. (2015) (https://doi.org/10.3390/w7010255) show that the selection of return periods for the loss

exceedance probability curve has a large effect on the EAD. Have you done any sensitivity analysis on how the selection of return periods is influencing your EAD estimates?

> Actually, each loss-exceedance probability curve comes from the catastrophe model part of the workflow so is based on 10,000 years of synthetic flood events with realistic spatial footprints. The return period maps are used to turn each of these spatially variable event intensity footprints into a composite flood depth map for which we can calculate a loss. This gives a distribution of losses with which to form a loss exceedance curve. The Expected Annual Damage is therefore just the integral of the loss exceedance curves. This approach differs significantly from the simpler method of calculating loss for a series of 'constant in space' return period maps and using these to compute an EAD as Olsen et al have done which. This is already discussed in the SI on lines 377-382. The choice of return periods in our method will somewhat influence the granularity with which footprints can be generated but the results are not expected to be significantly sensitive to this choice. Accordingly, the return periods were simply chosen to form a spread across the range of typical loss creating flood events and we will add some text to better explain this.
>
> We have added the following text at line 239-40:
>
> "Without such a stochastic method which includes spatial dependence it is only possible to compute Expected Annual Damage from a set of return period hazard layers."

P9 L249: You mention the "Fathom model" for the first time in the manuscript. I am assuming this is the name of the model you are presenting in the paper, but would be good to formally introduce the name to avoid confusion.

> "Fathom" was included in error, and we will remove this. The model was produced by Fathom (www.fathom.global) but as this is an academic work we did not want to be accused of advertising.
>
> "Fathom" has been deleted.

P9 L257: If possible, it would be great to have the equations for each metric in the text as it makes it easier for the reader to understand how those metrics are calculated.

> Of course. We will add these.
>
> The relevant equations have now been added.

Figure 2: Would be nice to have an inset showing the location of each flood layer on a GB/UK map

> We will try to do this. The tension here is that the plot is already a whole page figure so where to add an inset without reducing the size of each sub-panel (and hence losing detail) could be a problem. We will experiment with some potential solutions.
>
> We have now included a location inset on this figure.

Figure 2 caption: flood hazard maps on the right are shown in red not green

> Thanks! This is a mistake and will be corrected.
>
> Now corrected.

Table 2: Table 2 is an example, but comment is more general: it is sometimes not perfectly clear if values are for England, GB or the UK. As far as I am aware, the NaFRAs are conducted by each of the devolved nations individually. Is the number shown in Table 2, the sum of all NaFRAs or are these values for England only?

NaFRA is the name of the flood risk mapping programme in England only (although it did also cover Wales pre-2013).  Wales, Scotland and Northern Ireland have their own programmes with different methodologies and only report number of properties exposed and not financial losses.  To create a GB loss we therefore scale the NaFRA result for England using the ratios reported in Penning-Rowsell (2021). These were taken from the emulation methodology used in the 2017 UK Climate Change Risk Assessment (Sayers, 2017). This suggested that England accounts for 79% of flood losses, Scotland 12%, Wales 6% and Northern Ireland 2%.

You are right however that this is not very clear in the main text, and we will correct this. We have now made substantial changes to lines 368-379 to make this clear.

**Editor comments**

| Action | Response |
|---|---|
| (a) Figure 1 caption. You've used different colours and boxes that have heavy and non-heavy line widths. I suggest you refer to these in the caption so reader understands what the colours mean (or state, "See text for explanation of colours of boxes" which I think is a poorer solution, but viable if necessary). | Explanation now added to the figure caption. |
| (b) Figures. Some of your figures are not the most friendly for those with colour blindness, where colour is the only identifying difference (e.g., in Figure 1) for your narrative. You might want to consider running some of these through some simulators and/or picking more colour-blindness friendly palettes. | All figures have been reviewed and revised where necessary. Figure 1 in particular has been substantially changed. |
| (c) Space between units and numbers. Although this will be picked up by the copy editors, please put space between numbers and units (e.g., 15 cm, not 15cm). | Done |
| (d) Referring to URLs in the text. My suggestion is instead of referring to the URL in the text, you make this a full citation, and put in the reference list. For example, in reference list "NRFA (National River Flow Archive) (2022) The UK Gauging Station Network [Online] Available at: https://nrfa.ceh.ac.uk/uk-gauging-station-network [Last Accessed 26 November 2022]" (you can look online to recent NHESS articles for examples). And then in the text you would refer to "see NRFA (2022) for the river gauge network). | Done. I have left the links in for the footnotes of Table S1 as here they seemed to make sense. |
| (e) Table headers go above the tables. | Done |
| (f) In-text references. When you have more than one in-text citation in ( ), I see you've done alphabetically. My personal preference is from oldest to newest, but your choice here. | Have left this as is because this is what my reference manager software has for its EGU style sheet |
| (g) Figure 3. Although in most places n-dash vs. hyphen will be adjusted by copy editors, in Figure 3, you are using in your legend for (a) and axes for (b) hyphens where there should be n-dashes or a minus sign (twice length of a hyphen). Also, in the legend use an n-dash in the "-0.5 -- -0.25" (I've indicated the n-dash by -- here) which is | Changed |

| | |
|---|---|
| a little confusing with the minus sign. I suggest you do something like "--0.5 to --0.25" where -- is a minus sign (double the hyphen) so as to avoid confusion. | |
| (h) General (e.g., Figure 4). My preference in figure captions and table headers is to always make as self-standing as possible (in case removed from the text) and define (again) acronyms. Another example is Figure 6 caption, where you have EAD in the figure, but state Expected Annual Damage in the figure caption. Instead, you could just do "Expected Annual Damage (EAD)" in the figure caption. | Done |
| (i) Figure 5. Colour again. You've used three colours, green, orange, blue. The green dash is less thick than the blue dash, which is good, but between the orange and blue, they are almost the same thickness. Can these be made a bit more different (e.g., use dash dot for one of them, or use different thicknesses for all three). The basic idea is that colour alone does not define the line. | Figure now re-done with greater variation between these lines. |
| (j) General. For all figure captions/table headers, please go over them, and make as self-standing as possible. This is just a double check, as some seem fairly short, and might benefit by a few more words. | Done |
| (k) Supplementary Material. My understanding is this will not be formatted by Copernicus. Only if this were an appendix would it be formatted (and did you instead mean for this to be an appendix, which would then be part of the paper that is formatted?). Therefore, you'll want to ensure it looks good, 'as is', for what you upload. This means dealing with paragraphing, line numbers. The table S1 could look much more professional (remove all vertical an horizontal lines, leaving only the top and bottom horizontal line, and one below the header; single space instead of 1.5 spacing, repeat header row for when you go across multiple pages, etc.). Do same for other tables-- make them look a tad more 'final product'. Figure captions should appear on same page as the figures. N-dash vs. hyphens dealt with (e.g., ~20-25 m, should be ~20--25 m, where -- is an n-dash). Section numbers should all have S in front of them (e.g., S3.2, rather than 3.2). Figure S4--text is getting a bit too small to read here, so consider stacking one above another these figures or enlarging the text. Figure S5. This really does not work, with black text on dark grey background. I suggest you put a header for all pages. Consider each section starts on a new page (page break before). | All done, apart from the header which looked clunky with EGU page margins. |

---

## Author Response (AR2)

Professor PAUL BATES
B.Sc., Ph.D.
Professor of Hydrology

**SCHOOL OF GEOGRAPHICAL SCIENCES**
University of Bristol
University Road
Bristol
BS8 1SS
UK
Tel: +44 (0)117 928 9954
Fax: +44 (0)117 928 7878
E-mail: geog-office@bristol.ac.uk
www.ggy.bris.ac.uk

| | |
|---|---|
| *From*: | Professor Paul Bates |
| *Tel:* | +44-117-928-9108 |
| *E-mail:* | Paul.Bates@Bristol.ac.uk |

11th January 2023

Professor Bruce Malamud
Editor
Natural Hazards and Earth System Science

**Re. Author response for Bates et al.**

Dear Bruce,

Many thanks for the final editorial comments on this revised manuscript. We have implemented these and hope that the paper is now acceptable for publication.

Yours sincerely,

Paul Bates